# Preharvest Salicylate Treatments Enhance Antioxidant Compounds, Color and Crop Yield in Low Pigmented-Table Grape Cultivars and Preserve Quality Traits during Storage

**DOI:** 10.3390/antiox9090832

**Published:** 2020-09-06

**Authors:** María E. García-Pastor, Pedro J. Zapata, Salvador Castillo, Domingo Martínez-Romero, Daniel Valero, María Serrano, Fabián Guillén

**Affiliations:** 1Department of Food Technology, EPSO, University Miguel Hernández, Ctra. Beniel km. 3.2, 03312 Orihuela, Alicante, Spain; m.garciap@umh.es (M.E.G.-P.); pedrojzapata@umh.es (P.J.Z.); scastillo@umh.es (S.C.); dmromero@umh.es (D.M.-R.); daniel.valero@umh.es (D.V.); fabian.guillen@umh.es (F.G.); 2Department of Applied Biology, EPSO, University Miguel Hernández, Ctra. Beniel km. 3.2, 03312 Orihuela, Alicante, Spain

**Keywords:** anthocyanins, ripening, *Vitis vinifera*, postharvest, firmness

## Abstract

Previous reports reported on the effectiveness of preharvest salicylic acid (SA) treatment on increasing fruit quality properties although no information is available about acetyl salicylic acid (ASA) and methyl salicylate (MeSa) treatments. Thus, SA, ASA and MeSa were applied at 1, 5, and 10 mM in 2016 and at 1, 0.1 and 0.01 mM in 2017 to vines of ‘Magenta’ and ‘Crimson’ table grapes. Preharvest salicylate treatments at high concentration, 5 and 10 mM, delayed berry ripening and reduced crop yield, while ripening was accelerated and yield increased at lower concentrations. In addition, SA, ASA, and MeSa treatments, at 1, 0.1, and 0.01 mM, improved berry color due to increased concentration of total and individual anthocyanins, for both cultivars. Quality parameters, and especially, antioxidant bioactive compounds, such as total phenolics and total and individual anthocyanins, were found at higher levels in treated berries at harvest and during prolonged cold storage, the highest effects being found in 0.1 mM MeSa treated table grapes. Overall, it could be concluded that MeSa treatment at 0.1 mM could be the most useful tool to increase bioactive compounds with antioxidant properties in table grape and in turn, their health beneficial properties, with additional effects on increasing crop yield, accelerating on-vine ripening process and maintaining quality traits during prolonged storage.

## 1. Introduction

Table grape quality depends mainly on cluster size and shape, berry size, sugar/acidity ratio, aroma, and color. During grape development, veraison is considered the most important stage since most of the changes associated with maturation usually start, such as sugar accumulation, acidity reduction, onset of pigment occurrence and synthesis of volatile aroma compounds. These quality traits are going on until berry reaches full maturity [1]. In addition, grapes contain bioactive antioxidant compounds, such as vitamins and phenolic compounds, which have beneficial effects on human health, namely anti-inflammatory, anticancer, and anti-diabetic effects as well as effect on preventing cardiovascular diseases [2,3,4], which could depend on the gut microbiota composition [5]. Among phenolics, anthocyanins have special importance since they are responsible for color of all red, purple, and dark-purple *Vitis vinifera* L. cultivars, either destined to fresh consumption or winemaking [1,4]. However, ‘Magenta’ and ‘Crimson’ are red and purple skin table grape cultivars, respectively, with low color development. Moreover, berry pigmentation in the cluster of these cultivars is heterogeneous, which leads to depreciation of their market value. These problems are attributed to high temperatures in the Southeast of Spain during berry ripening, which do not allow proper color development [6]. 

To improve berry coloration, some attempts were performed in recent years. Thus, ethephon (an ethylene- releasing compound) and abscisic acid (ABA) treatments at veraison stage increased skin anthocyanin concentration although most of these studies were performed with wine grape cultivars [1]. In particular, in ‘Crimson Seedless’, treatments of vines with ABA or sucrose improved berry coloration [6,7] as well as regulated deficit irrigation applied at post-veraison stage [8]. However, the effects of ethephon on color development are inconsistent and can cause berry softening and the high cost of ABA reduces its practical application. Thus, new research are needed to find out other cost-efficient preharvest treatments able to induce anthocyanin biosynthesis in table grape. Recently, it was reported that Methyl jasmonate (MeJa) treatments at 0.1 mM, applied at key points of berry development, accelerated color evolution due to increased anthocyanin biosynthesis in ‘Magenta’ and ‘Crimson’ table grape cultivars [9].

Salicylic acid (SA) and its derivatives, acetylsalicylic acid (ASA) and methyl salicylate (MeSa) are presently considered hormonal compounds with a wide range of physiological effects on plant tissues, from germination to flowering, although the most studied ones are the induction of systemic acquired resistance and resistance against abiotic stresses [10]. In this sense, postharvest treatments with salicylates reduced chilling injury and decay (by increasing fruit resistance to diseases) in a wide range of commodities, with also positive effects on improving fruit quality properties, such as appearance, texture, and nutritional content [10,11,12]. In addition, in recent years, preharvest treatments with salicylates were reported to improve fruit quality attributes and fruit resistance to pathogen attacks. Thus, foliar spray treatments of jujube plants with 2 mM SA decreased decay caused by *Alternaria alternata* and *Monilinia fructicola*, either at harvest or during cold storage [13]. SA treatment of wax apple fruit, 24 h before harvesting, maintained fruit firmness and visual appearance during storage and higher levels of phenolic content and antioxidant enzyme activities [14]. Moreover, SA, ASA, or MeSa treatments of sweet cherry trees, at 0.5, 1 and 2 mM concentrations, applied at key points of fruit development increased fruit quality attributes at harvest, such as weight, firmness and content of bioactive compounds namely phenolics, including anthocyanins, which were maintained during storage [15,16,17]. Similar effects of salicylate preharvest treatments were reported in plum [18,19]. In addition, SA, ASA and MeSA are natural compounds in plants which are recognized as GRAS (generally recognized as safe) for the United States Food and Drug Administration (FDA) and previous reports showed that fruit treatments with salicylates do not impart taste or off-flavor to fruit, although some sensory attributes, such as sweetness and firmness, increased [15,16,17,18,19]. 

In table grape, there are a few recent reports regarding the effects of SA preharvest treatments on berry quality. Thus, Champa et al. [20] showed that 1.5 and 2 mM SA treatments of ‘Flame Seedless’ cultivar maintained berry color, firmness, phenolic content, and organoleptic properties during cold storage. Similar results were reported by Lo’ay and EL-Boray in SA-treated table grapes of this cultivar during storage at 28 °C [21]. Accordingly, preharvest SA treatments reduced berry weight loss and softening during storage at ambient temperature in ‘Thompson’ table grape cultivar [22]. In the white seeded ‘El-Bayadi’ cultivar, preharvest SA treatment led to grapes with higher total antioxidants (TA) and total phenolic and flavonoids content at harvest, which was attributed to a delay of the on-plant ripening process [23]. Moreover, SA treatment at pre-veraison stage of ‘Sahebi’ grapes led to increases not only in total phenolics and flavonoids at harvest but also in anthocyanin concentration, especially in malvidin-3-glucoside, the major anthocyanin in this cultivar [24]. According to these previous reports, we hypothesized that anthocyanin content would be increased by salicylate preharvest treatments in ‘Magenta’ and ‘Crimson’ table grapes cultivars, improving their color development and in turn, their market value as well as their antioxidant properties. Thus, the effects of SA, ASA, and MeSa preharvest treatments on berry quality properties at harvest and during cold storage were evaluated in these two table grape cultivars over two growing seasons, with special emphasis in color and anthocyanin content. This is worth noting that as far as we know, there are not previous reports regarding ASA and MeSa treatments on any grape cultivar either applied as pre- or post-harvest treatment.

## 2. Materials and Methods 

### 2.1. Plant Material, Treatments and Experimental Design

This study was performed with ‘Magenta’ and ‘Crimson’, two different *Vitis vinifera* L. seedless table grape cultivars, both of them grafted onto Paulsen 1103 rootstocks, during two growing seasons (2016 and 2017). Vines were planted in a sandy soil, at 2.5 × 3 m, in a commercial orchard in Calasparra (Murcia, Spain). The training system consisted on an overhead trellis at a height of 1.9 m above ground level which was upper and laterally covered with a thread warp net to protect the vines from hail, birds, and insects. Vines were irrigated according to their water requirements along the growth cycle by using a programmer drip irrigation system consisting on a drip irrigation line per row with three emitters per plant. Fertilizers were applied in the irrigation system and pruning and thinning were carried out according to local cultural practices for table grape. SA, ASA, and MeSa treatments were applied at 1, 5 and 10 mM concentrations in 2016 and at 1, 0.1 and 0.01 mM concentration in 2017, since in general, salicylates at 5 and 10 mM decreased vine yield and delayed berry ripening process, as commented in Result section. Treatments were performed by spraying 1.5 L per vine of freshly SA, ASA, or MeSa (purchased from Sigma-Aldrich, Darmstadt, Germany) prepared solutions containing 0.5% Tween 20 as surfactant. The treatments were carried out at early morning and under favorable weather conditions in which no rain or winds were forecast for the next 24 h. An aqueous solution of 0.5% Tween was used to spray control vines. Three treatments were applied for each compound: when berries had ca 40% of their final volume (≅1650 and 2100 mm^3^ for ‘Crimson’ and ‘Magenta’, respectively), at veraison stage and 3 days before the first harvest date (Table 1). Cultural practices during the experiments, such as pruning, irrigation and fertilization, followed standard procedures for table grape crop. A completely randomized block design by using three replicates of three vines for each treatment, cultivar and year was set up.

### 2.2. Vine Yield Determination and Storage Experiment

Clusters were harvested when berries reached the commercial ripening stage according to size, color and total soluble solid content of these cultivars, 170–180 g·100 g^−1^. Then, four harvests were performed for both cultivars and years, and kg harvested from each vine was measured for each harvest date. Production for each vine was expressed as accumulated yield (kg·vine^−1^) for the first to the last harvest date (mean ± SE of three replicates of three vines).

In 2016, storage experiment was performed with table grapes from control and 1 mM SA, ASA and MeSa treated vines of the second harvest date (27th July and 3rd August for ‘Magenta’ and ‘Crimson’, respectively), while in 2017, storage experiment was performed with table grapes from control and SA, ASA, MeSa treated table grapes at 1, 0.1 and 0.01 mM of the second harvest date (31st July and 10th August for ‘Magenta’ and ‘Crimson’, respectively). Storage experiments were performed when enough number of clusters were harvested which was at the second harvest date for both years and cultivars. In both years, 8 clusters from each replicate and cultivar of control and treated vines were immediately transported to the laboratory and stored at 2 °C and 90% RH for 0, 15, 30 and 45 days. For each sampling date, two clusters were taken at random from each replicate (three biological replicates each of one consisting on two clusters) in which the following parameters were measured. 

### 2.3. Quality Parameters

Clusters were weighed at day 0 and after each storage period, and weight loss was expressed in percentage with respect to weight at harvest. Color L*, a* and b* parameters were measured individually in 30 berries from each replicate (15 berries from each of the two clusters per replicate) by using a Minolta colorimeter (CRC200, Minolta Camera Co., Osaka, Japan), and color was expressed as Hue angle (arctan b/a). Firmness was measured as the force that achieved a 5% deformation of the berry diameter by using a Texture Analyzer (TX-XT2i, Stable Microsystems, Godalming, UK), and was expressed as N·mm^−1^. Data of color and firmness are the mean ± SE of three replicates of 30 berries. After that these 30 berries of each replicate were cut in small pieces and ground to obtain a homogeneous juice sample. Total soluble solids (TSS) were measured in duplicate in each juice sample by using a digital refractometer (Atago PR-101, Atago Co. Ltd., Tokyo, Japan) at 20 °C, and expressed as g·kg^−1^ on a fresh weight (FW) basis (mean ± SE). Total acidity was measured, also in duplicate, in the same juice by automatic titration with 0.1 N NaOH up to pH 8.1 (785 DMP Titrino, Metrohm, Phyathai, Thailand) and results (mean ± SE) are expressed as g tartaric acid equivalent kg^−1^ FW. 

### 2.4. Total Phenolics and Total and Individual Anthocyanin Quantifications

Another sample of 30 berries from each replicate was taken as above, cut in small pieces, ground under liquid N_2_ and stored at −80 °C until quantification of total phenolics and total and individual anthocyanins were performed. Phenolic compounds were extracted by homogenizing 5 g of grape sample with 10 mL of water: methanol (2:8, *v/v*) containing 2 mM NaF for 30 s by using a homogenizer (Ultraturrax, T18 basic, IKA, Berlin, Germany). The extracts were centrifuged at 10,000× *g* for 10 min at 4 °C and total phenolics were quantified in the supernatant (in duplicate in each extract) by using the Folin-Ciocalteu reagent according to previous report [19]. Anthocyanins were extracted by homogenizing manually 10 g of frozen berry tissues with 15 mL of methanol/formic acid/water (25:1:24, *v/v/v*) by using a mortar and pestle. Then, the homogenate was sonicated in an ultrasonic bath for 60 min and then centrifuged at 10,000× *g* for 15 min. To measure total anthocyanin concentration in the supernatant (in duplicate) the absorbance at 520 nm was read in an spectrophotometer (UNICAM Helios-α, Artisan Technology Group, Champaign, IL, USA) and results were expressed as g of malvidin 3-glucoside equivalent (molar absorption coefficient of 27,000 L·cm^−1^·mol^−1^ and molecular weight of 493.4 g·mol^−1^) per kg FW (mean ± SE). For individual anthocyanin quantification, the supernatant was filtered through a 0.45 μm fluoruro de polivinilideno (PVDF) filter (Millex HV13, Millipore, Bedford, MA, USA) and 20 µL were injected into a high-performance liquid chromatography (HPLC) system (Agilent HPLC1200 Infinity series, Agilent Technologies Inc., Waldbronn, Germany) working as previously reported [19]. Chromatograms were recorded at 520 nm. Anthocyanin standards were: malvidin 3-glucoside and peonidin 3-glucoside (purchased from Sigma-Aldrich, Darmstadt, Germany) and cyanidin 3-rutinoside (purchased from Polyphenols SA, Sandnes, Norway) and results (g·kg^−1^ FW) were mean ± SE. Delphinidin 3-glucoside and petunidin 3-glucoside were expressed as malvidin 3-glucoside equivalents. 

### 2.5. Statistical Analysis

Data were subjected to ANOVA analysis, with treatments and storage time as sources of variation. Tukey’s test was used to examine whether mean differences were significant at *p* < 0.05. All analyses were performed by using SPSS software package v. 11.0 for Windows (SPSS, 2001, IBM Corporation, Armonk, NY, USA). In addition, a Student *t*’ test was performed when comparing data of two sampling dates or years.

## 3. Results

### 3.1. Vine Yield and Berry Ripening Process

Clusters were harvested when berries reached their commercial ripening stage based on skin color (homogeneous purple color of the berries in a cluster) and TSS content characteristic of these cultivars (170–180 g·100 g^−1^). Given the heterogeneous ripening of the clusters within a vine, four harvests were performed for both table grape cultivars and years. Total yield of vines of ‘Magenta’ cultivar treated with SA or ASA at 5 and 10 mM was significantly decreased (*p* < 0.05) with respect to yield of control vines in 2016 experiment. Thus, total yield was 34.18 ± 1.34 kg·vine^−1^ in control vines, 26.40 ± 4.66 and 18.90 ± 4.85 kg·vine^−1^ in those treated with SA at 5 and 10 mM, respectively, and 18.91 ± 4.89 and 20.77 ± 1.60 kg·vine^−1^ in 5 and 10 mM ASA treated ones, respectively, while no significant differences were observed between 1 mM SA or ASA treatments and controls (Appendix A). However, for MeSa treatments different results were obtained since total yield was not significantly (*p* > 0.05) affected by any of the MeSa doses. In addition, for the first, second and third harvest dates, more kg·vine^−1^ were harvested from MeSa treated vines than from controls, especially for 1 mM dose (Appendix A). Thus, given the results obtained in year 2016, vines of ‘Magenta’ cultivar were treated with SA and its derivatives at 1, 0.1 and 0.01 mM concentrations for 2017 experiment. Results from salicylate treatments at 1 mM concentration confirmed those of 2016 experiment. However, SA and ASA treatments at 0.1 and 0.01 mM increased significantly (*p* < 0.05) the accumulated vine yield for all harvest dates, as well as MeSa at 0.1 and 1 mM (Appendix A), showing an effect on accelerating the on-vine berry ripening process.

In ‘Crimson’ cultivar results were different in some extension because the delay of the on-vine ripening process and the reduction on total yield was only evident and significant (*p* < 0.05) for SA and ASA at 10 mM but not for 5 mM as occurred in ‘Magenta’ cultivar. Thus, total yield was 43.77 ± 4.36 kg·vine^−1^ in controls and 35.42 ± 2.07 and 27.56 ± 4.29 kg·vine^−1^ in 10 mM SA and ASA treated ones, respectively (Appendix A). With respect to MeSa treatments, 10 mM dose significantly (*p* < 0.05) reduced total yield which was 37.48 ± 3.16 kg·vine^−1^, (14.37% less than control) (Appendix A). For 2017 experiment 1, 0.1 and 0.01 mM of SA, ASA and MeSa were applied. Total yield was not significantly affected by any of the salicylate treatments, although more kg·vine^−1^ were harvested the first and second harvest dates from vines treated with 1 mM of SA and ASA (Appendix A) confirming the results of 2016 experiment. For MeSa treatments at 1, 0.1 and 0.01 mM concentrations the amount of kg harvested the first and second harvest dates was significantly higher (*p* < 0.05) in treated vines than in controls, the highest effect being found with 0.1 mM concentration (Appendix A), showing that the on-vine ripening process was accelerated.

### 3.2. Bioactive Compounds: Anthocyanins and Phenolics

In 2016 experiment, total anthocyanin concentration at harvest was significantly higher (*p* < 0.05) in grapes from 1 mM SA, ASA and MeSa treated vines than in controls in both cultivars and increases occurred during storage in all of them, although anthocyanin concentration was maintained at higher levels in treated berries, the highest increases being found for 1 mM MeSa treatment in both cultivars (Figure 1A,B). 

Total anthocyanin concentration at harvest was also significantly (*p* < 0.05) increased by preharvest salicylate treatments in 2017 experiment in both cultivars, and the highest concentrations were found for 0.01 mM SA, 0.1 mM ASA and 0.1 mM MeSa treatments in ‘Magenta’ (Figure 2A) and for 0.01 mM SA, 1 mM ASA and 0.1 mM MeSa in ‘Crimson’ (Figure 2B). Thus, total anthocyanin content during storage was measured in table grapes from these treatments and results showed increases during storage, as occurred in 2016 experiment, with higher concentrations in berries from treated vines than in those from control ones until the last sampling date (Figure 2C,D).

Individual anthocyanins were quantified in grape samples from control and 1 mM SA, ASA and MeSa treated vines in 2016 experiment and in samples from control and vines treated with the best concentration of SA, ASA, and MeSa in terms of their effects on increasing total anthocyanin concentration in 2017 experiment. Thus, five individual anthocyanins were identified and quantified in ‘Magenta’ cultivar in both years, the major ones being peonidin 3-*O*-glucoside (Pn-3-glu) and malvidin 3-*O*-glucoside (Mv-3-glu), with concentrations between 0.015 and 0.017 g·kg^−1^, followed by delphinidin 3-*O*-glucoside (Dlp-3-glu), ca. 0.006 g·kg^−1^, and petunidin 3-*O*-glucoside (Pt-3-glu) and cyanidin 3-*O*-glucoside (Cy-3-glu) at very low concentrations, between 0.002 and 0.003 g·kg^−1^ in table grapes from control vines (Figure 3A,C). For ‘Crimson’ table grapes, only three anthocyanins were detected, the major one being Pn-3-glu, followed by Mv-3-glu and Cy-3-glu, with concentrations ca. 0.04–0.042, 0.005–0.006 and 0.001 g·kg^−1^, respectively in control berries (Figure 3B,D). In 2016 experiments, salicylate treatments led to berries with significant (*p* < 0.05) higher concentration in all the individual anthocyanins, especially in the major ones, and the highest increases were found for 1 mM MeSa treatment in both cultivars (Figure 3A,B). Similar effects of salicylate treatments on increasing individual anthocyanin concentration were observed in both table grape cultivars in 2017 experiment, the major increase being found for 0.1 mM MeSa treatment (Figure 3C,D).

Total phenolic concentration at harvest in 2016 was significantly higher (*p* < 0.05) in berries from salicylate treated vines than in controls, the effect being higher for MeSa treatment, followed by ASA and SA ones in ‘Magenta’ (Figure 4A), while no significant differences were found among salicylate treatments in ‘Crimson’ (Figure 4B). Phenolic concentration increased during storage although they were maintained at higher levels (*p* < 0.05) in treated than in control berries, the higher effects being found in MeSa treatment for both cultivars. Similarly, higher total phenolic concentrations were found in treated than in control berries at harvest in 2017 experiment, especially for 0.1 mM MeSa in ‘Magenta’ (Figure 4C) and 0.01 mM SA in ‘Crimson’ (Figure 4D).

Total phenolic concentration increased during storage in berries from control and treated vines, as was found in 2016 experiment, although values were significantly (*p* < 0.05) lower in control berries than in treated ones (Figure 5A,B). The highest effect on increasing total phenolic content during the whole storage period was found for 0.1 mM Mesa and 0.01 mM SA treatments in ‘Magenta’ and ‘Crimson’ cultivars, respectively. 

### 3.3. Quality Parameters

Quality parameters of table grapes, such as color, TSS, TA, firmness and weight loss were measured at harvest and during storage. Values of Hue angle colour were significantly lower (*p* < 0.05) in 1 mM salicylate treated grapes than in controls at harvest in 2016 experiment (Figure 6A,B), showing that treated berries had a deeper purple colour than control ones, the highest effect being found in 1 mM MeSa treated berries for both cultivars. Values of Hue angle decreased during storage in berries from control and treated vines for both cultivars, although they were always higher in control than in treated ones. In 2017 experiment, when SA, ASA and MeSa were applied at 1, 0.1 and 0.01 mM concentrations, ‘Magenta’ and ‘Crimson’ berries had also significant (*p* < 0.05) lower values of Hue angle colour than control ones at harvest and decreased occurred during storage (Appendix A). The highest effect on decreasing Hue angle among the three assayed concentration was found at 0.01 mM for SA treatments (Appendix A) and for 0.1 mM for MeSa treatments (Appendix A) in both cultivars, while for ASA treatments, the highest effect was found for 0.1 mM concentration in ‘Magenta’ (Appendix A) and for 1 mM in ‘Crimson’ (Appendix A). 

Concentrations of TSS at harvest were 195.5 ± 1.0 and 180.2 ± 2.2 g·kg^−1^ in ‘Magenta’ berries and 183.0 ± 2.5 and 175.8 ± 1.2 g·kg^−1^ in ‘Crimson’ for 2016 and 2017, respectively and they generally increased during storage. However, TSS were significantly (*p* < 0.05) higher in 1 mM SA, ASA and MeSa treated berries than in controls, at harvest and after 45 days of cold storage, while no significant differences were found for 0.1 and 0.01 mM salicylate treatments (Table 2). On the contrary, TA values decreased during storage in control and treated table grapes of both cultivars, although values were higher in treated than in controls, the effects being significant (*p* < 0.05) for all the applied doses (Table 3). Berry firmness also decreased during storage and values were significantly (*p* < 0.05) higher in salicylate treated table grapes than in control, at harvest and during cold storage for both cultivars and years. In 2016 experiment, no significant differences were observed among 1 mM SA, ASA and MeSa treatments on ‘Magenta’ while in ‘Crimson’ the highest firmness values were observed for 1 mM SA treated berries (Appendix A). In 2017 experiment, the highest values of firmness, either at harvest or for each sampling date during storage, were observed for 0.01 mM SA dose and 0.1 mM MeSa dose in both cultivars and for 0.1 and 1 mM ASA doses for ‘Magenta’ and ‘Crimson’ cultivars, respectively (Appendix A). Finally, cluster weight losses increased during storage reaching final values of 7.13 ± 0.24%and 9.09 ± 0.36% in ‘Crimson’ and ‘Magenta’ control table grapes, respectively, in 2016 experiment and significantly (*p* < 0.05) lower values in grapes from salicylate treated vines (Appendix A). Accordingly, weight losses during storage were also reduced in 2017 experiment as a consequence of preharvest salicylate treatments, the highest effects being found for 0.1 mM MeSa treatment in both cultivars (Appendix A).

## 4. Discussion

Results demonstrated that treatments with salicylates affected vine yield and grape ripening depending on the applied compound, concentration, and cultivar. Grapevine treatment with SA or ASA at 5 and 10 mM led to a significant decrease (*p* < 0.05) on total yield of ‘Magenta’ while these effects only were significant for 10 mM dose on ‘Crimson’ in 2016 experiment. No berry drop was observed as a consequence of salicylate treatments and morphological traits of clusters was similar in control and treated vines. Thus, the effects of salicylate treatments at high doses on reducing total yield were attributed to a delay or inhibition on the ripening process since many berries failed to ripen properly and some clusters did not reach the requested commercial quality and were discarded. However, when lower doses of SA and ASA were applied, the ripening process was accelerated, as was observed for SA and ASA at 0.1 and 0.01 mM in ‘Magenta’ and at 1 mM in ‘Crimson’. On the contrary, all the applied MeSa concentration, except 0.01 mM, accelerated the ripening process in ‘Magenta’ and except 10 mM in ‘Crimson’, although for both cultivars the highest effect was found for 0.1 mM dose. Thus, to accelerate the ripening process and achieve higher prices at market the most appropriate concentration of each salicylate should be established for each particular table grape cultivar. In fact, different effects of preharvest SA treatments on grape ripening were reported depending on cultivar and applied concentration. Thus, in ‘Flame Seedless’ preharvest treatments with 1.5 and 2 mM of SA hastened berry ripening but 1 mM has not effect [20], while ripening was delayed by 0.72 mM SA in ‘Thompson’ grapes [22]. Accordingly, ripening also was delayed in the wine grape cultivar ‘Syrah’ by 0.72 and 3.6 mM SA foliar spray treatments at veraison stage [25]. Similar results were obtained by preharvest treatment with 4 mM SA on the white table grape cultivar ‘El-Bayadi’ [23] and by 7.2 mM SA injected into berries before veraison [26], the delay being attributed to the antagonist effects of SA on ABA biosynthesis, which is the main hormone implied in the ripening of this non-climacteric fruit. In general, these results and the present ones show that preharvest salicylate treatments at high concentration led to a delay the ripening process in grapes while it could be hastened by lower doses. Accordingly, in other non-climacteric fruit, such as sweet cherry, SA and ASA at 2 mM delayed the on-tree ripening process while not effect was observed for 0.5 and 1 mM concentrations [15].

On the other hand, increases in productivity by SA treatments were reported in other crops due to enhanced leaf area, photosynthetic pigments concentration in leaves and photosynthesis rate [27]. However, these effects depend on applied dose. For instance, treatment of cucumber plants with 0.075, 0.1 and 0.15 mM led to increased fruit weight and crop yield while they were reduced by 0.25 and 0.5 mM SA treatments [28]. Accordingly, the present results show different effects of salicylate treatments on crop yield depending not only on applied doses but also on cultivar. Thus, in ‘Magenta’ cultivar, SA, and ASA at 0.1 and 0.01 mM and MeSa at 0.1 mM increased total yield significantly (*p* < 0.05) while it decreased by 5 and 10 mM SA and ASA doses. However, in ‘Crimson’ total yield was significantly reduced by SA and ASA at 10 mM but not significantly enhanced by any of the salicylate treatments (Appendix A). The increase in yield in ‘Magenta’ was due to an increase on berry volume of 5%, 7.6% and 13% as a consequence of 0.01 mM SA, 0.1 ASA and 0.1 MeSa treatments, respectively. Accordingly, previous reports showed an increase on fruit size on sweet cherry and plum after SA, ASA, or MeSa preharvest treatments [15,16,18] and on pepper fruit from plants treated with MeSa [29], which were attributed to an increase on sugar translocation from leaves to fruit. It is worth noting that as clusters with higher berry size are more appreciated by consumers and reach higher prices at markets than small ones, these treatments could increase economic benefit of this crop, apart from their effects on increasing total yield. In other table grape cultivars, it was reported that 1, 1.5, and 2 mM SA increased cluster size and yield on ‘Flame Seedless’ [20] as well as 0.72 mM SA in ‘Thompson’ [22]. However, results of the present research show that lower SA concentration could be even more effective to increase yield in the ‘Magenta’ cultivar, although it is not applicable to all table grape cultivars because no significant increases on yield were observed in ‘Crimson’ for the wide range of SA doses assayed.

Results of color hues show that salicylate treatments improved color in ‘Magenta’ and ‘Crimson’ table grapes, since the lower values of Hue angle obtained in treated berries, either at harvest or during storage, show deeper red and purple colors which were due to increases in anthocyanin biosynthesis. In fact, highly negative correlations were found between anthocyanin concentration and color Hue values for both table grape cultivars and years taking into account data of control and treated berries for all sampling dates during storage (‘Magenta’ 2016: y = −0.0068x + 0.1515, r^2^ = 0.9720; ‘Crimson’ 2016: y = −0.0033x + 0.1338, r^2^ = 0.8567; ‘Magenta’ 2017: y = −0.0103x + 0.2283, r^2^ = 0.9059; ‘Crimson’ 2017: y = −0.0085x + 0.2137, r^2^ = 0.9257). Thus, salicylate treatments would lead to improve the market quality of these cultivars, usually depreciated by their lack of proper coloration. The effects of SA, ASA, and MeSa preharvest treatments on increasing anthocyanin content was reported in sweet cherry [15,16,17] and plum [18,19]. In grape no previous reports about ASA or MeSa treatments are available in the literature, although a few ones regarding SA treatments were reported. Oraei et al. [24] reported that SA spraying treatment, at concentrations from 50 to 200 mM on “Sahebi” cultivar, at pre-veraison stage, led to an increase in phenolic and anthocyanin content. The authors explained these results as the effect of an activation of phenylalanine ammonia lyase (PAL) activity in the vine. Chen et al. [30] reported that in vivo infiltration of 150 μM SA into entire ‘Cabernet Sauvignon’ berries after harvest activated PAL by enhancing the accumulation of PAL mRNA and the synthesis of a new PAL protein as well as the enzyme activity. In addition, it was reported in Chinese cabbage that SA increases the expression of genes codifying by enzymes such as chalcone synthase (CHS) and chalcone isomerase (CHI), which are involved further downstream in the pathway of flavonoids [31]. These effects of salicylate treatment on increasing PAL activity would be also responsible for the enhanced total phenolic concentration found in berries from treated vines. Thus, salicylate preharvest treatments would lead to increase antioxidant properties and health beneficial effects of table grape consumption given the recognized role of phenolic including anthocyanins in health beneficial properties [2,3,4,25,31]. These effects would be even higher after prolonged cold storage since, in general, the highest differences among control and treated berries in total phenolic and total anthocyanin concentrations were found at the last sampling date. As a general trend, 1.5–2 folds’ increases were found in total phenolic and anthocyanin concentration from harvest to day 45 of cold storage, which cannot be attribute to concentration of the compounds in berry tissues due to weight losses because were lower than 10% in both cultivars and years. Increases in total phenolic and anthocyanin concentrations during cold storage were reported in other table grape cultivars, such as ‘Flame Seedless’ and ‘Red Globe’ after 45 and 60 days of storage, respectively, in which weight losses were ca. 1% [32,33].

On the other hand, previous studies showed that climatic conditions, especially high temperatures, have a detrimental effect on color and anthocyanin accumulation in grapes from veraison to ripening [34,35]. In our experimental conditions, medium and maximum temperatures from July to September, when veraison and ripening occurred in both table grape cultivars, were similar for 2016 and 2017 (Appendix A). However, minimum temperatures of July, August and September were lower in 2017 than in 2016 (Appendix A) which was related to a higher content on total anthocyanins in control and 1 mM salicylate-treated berries for both cultivars in 2017 experiment (Appendix A). These results could be attributed to both lower expression levels of anthocyanin biosynthetic genes and lower activities of anthocyanin biosynthetic enzymes, particularly UDP-glucose flavonoid 3-*O*-glucosyltransferase (UFGT, a key enzyme in anthocyanin biosynthesis), as reported by Mori et al. [34]. With respect to total phenolic concentration at harvest, higher values were found in control and 1 mM salicylate-treated berries in 2016 than in 2017 for ‘Crimson’ cultivar, while the contrary occurred in ‘Magenta’ (Appendix A) and in turn a clear relationship with the temperatures from version to ripening cannot been found. In fact, in a recent study with wine grape cultivars, it was reported that the impact of climatic variables on phenolic content is very complex, since maximum, minimum and medium temperatures as well as rain and water stress along the growing cycle have a different impact on individual phenolic compounds [36] and then further research is required to better understand these relationships.

With respect to quality parameters, increases in weight loss and TSS and decreases in TA and firmness occurred during cold storage in all berries which are related to the postharvest ripening process in fresh fruit, including table grape, and lead to quality deterioration and losses of their marketable value [37,38,39]. However, the evolution of these quality parameters was delayed in salicylate-treated grapes of both cultivars. Accordingly, SA preharvest treatment of ‘Thompson’ (at 0.72 mM, at pea and veraison stages) reduced softening and weight loss during storage at 20 °C [22], as well as 2 and 4 mM SA treatments (at veraison stage and 14 days before harvesting) reduced cluster water loss, rachis browning index and softening during storage at 28 °C in ‘Flame Seedless’ grapes, due to a reduced activity of the enzymes polygalacturonase, xylanase and cellulase [21]. The preservation of quality parameters and organoleptic properties during cold storage were also reported by Champa et al. [20] in ‘Flame Seedless’ cultivar as a result of preharvest treatments with SA (1.5 and 2.0 mM) at pea stage and at veraison. As postharvest treatment, SA (1, 2, and 4 mM) improved berry and cluster appearance during storage for up to 45 days at 0 °C, followed by 2 days at 20 °C, on ‘Bidaneh Ghermez’ grapes [40]. Moreover, in ‘Flame Seedless’ table grape cultivar, it was reported that the combination of pre- and post-harvest SA treatment was even more effective on maintaining grape quality than pre- or postharvest treatment alone, since higher firmness, lower weight loss and better appearance of berries as well as of rachis were observed after long storage time [41]. According to the present results and the commented previous ones, it is clear that preharvest salicylate treatments could be considered an effective tool to maintain table grape quality during storage throughout delaying the postharvest ripening process. However, the mechanism involved is still unclear although it could be related to the increase of the SA endogenous levels induced by salicylate treatment. In this sense, Zhang et al. [42] showed that kiwifruit ripening process was correlated with a decrease in SA endogenous concentration, while ASA treatment increased endogenous levels of SA and delayed ripening and senescence, manifested by lower softening, lipoxygenase activity and reactive oxygen species (ROS) production. Moreover, in plum and sweet cherry, it was reported that the activity of the antioxidant enzymes superoxide dismutase (SOD), peroxidase (POD), ascorbate peroxidase (APX) and catalase (CAT) increased by preharvest treatments with SA, ASA, and MeSa [17,18,43]. Taking into account that ROS production in increased during the postharvest ripening process, the increase in the activity of these antioxidant enzymes, together with enhanced concentrations of antioxidants compounds such as anthocyanins and phenolics, would lead to a more efficient system of ROS cleaning and in turn, to delay berry ripening and senescence processes, being responsible for the maintenance of berry quality attributes during prolonged cold storage found in table grapes from salicylate-treated vines.

## 5. Conclusions

Results show that the effect of SA, ASA, and MeSa preharvest treatments on yield and quality attributes of table grape at harvest and during storage depends on applied compound, concentration, and cultivar. However, considering the overall results, it could be concluded that 0.1 mM MeSa treatment could be a useful tool to increase crop yield and accelerate on-vine ripening process on both cultivars, which would lead to improve the economic profit of this crop. In addition, this treatment was the most effective on enhancing anthocyanin biosynthesis and berry color in these poorly colored cultivars. Moreover, quality parameters of grapes from treated vines were maintained during cold storage at higher levels as compared with those from controls. It is worth noting that total anthocyanins and phenolics, which have antioxidant properties, were found at higher concentrations at harvest and during prolonged cold storage in treated berries, which would lead to increase the health beneficial effects of table grape consumption. In addition, SA, ASA, and MeSa are natural compound, present in almost all plant tissues that have always been consumed by humans so it does not imply adverse effects on human health.

## Figures and Tables

**Figure 1 antioxidants-09-00832-f001:**
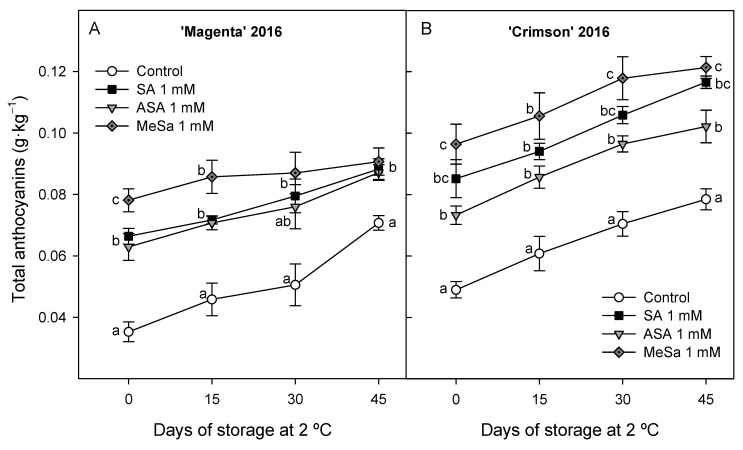
Effects of preharvest 1 mM salicylic acid (SA), acetyl salicylic acid (ASA) and methyl salicylate (MeSa) treatments on total anthocyanin concentration on ‘Magenta’ (**A**) and ‘Crimson’ (**B**) cultivars during storage at 2 °C in 2016 experiment. Data are the mean ± SE of quantifications made in duplicate in three replicates. Different letters show significant differences (*p* < 0.05) among treatments for each sampling date.

**Figure 2 antioxidants-09-00832-f002:**
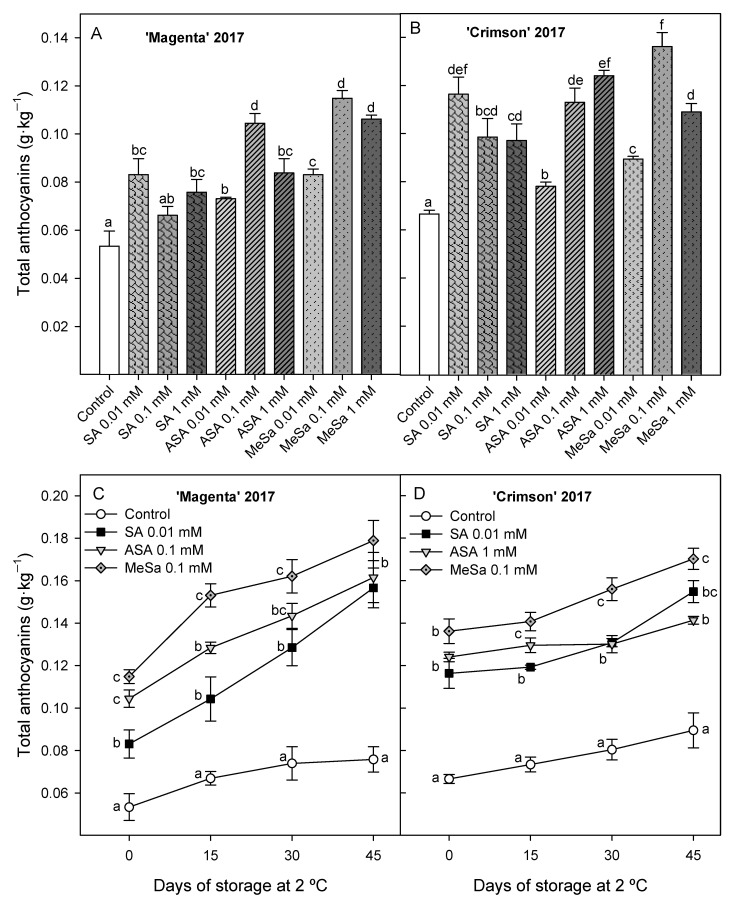
Total anthocyanin concentration at harvest in control and salicylic acid (SA), acetyl salicylic acid (ASA) and methyl salicylate (MeSa) treated berries of ‘Magenta’ (**A**) and ‘Crimson’ (**B**) cultivars and during storage (**C**,**D**) at 2 °C in 2017 experiment. Data are the mean ± SE of quantifications made in duplicate in three replicates. Different letters show significant differences (*p* < 0.05) among treatments for each sampling date.

**Figure 3 antioxidants-09-00832-f003:**
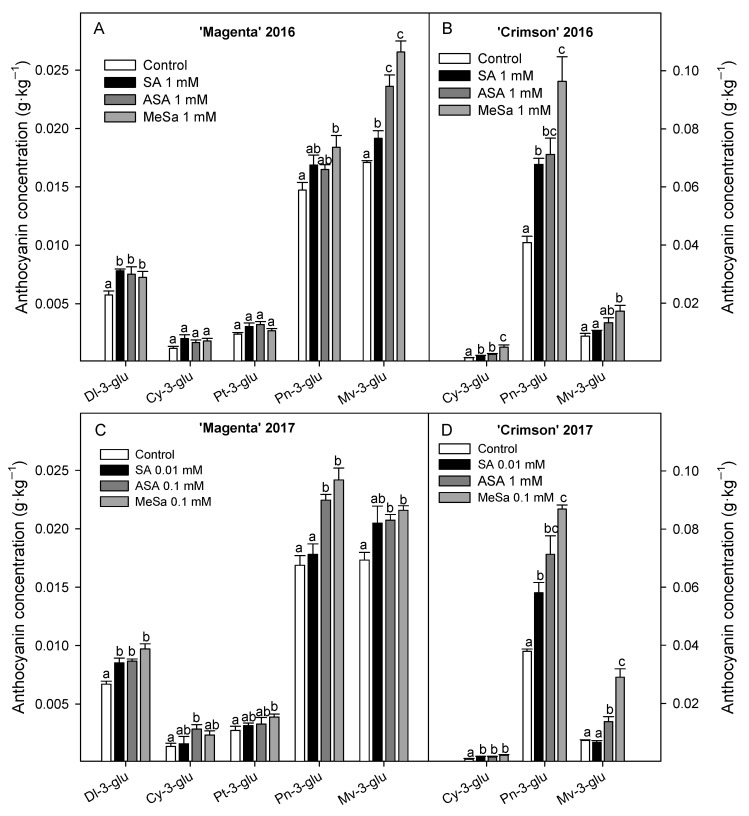
Individual anthocyanin concentration in ‘Magenta’ and ‘Crimson’ table grapes as affected by salicylic acid (SA), acetyl salicylic acid (SA) and methyl salicylate (MeSa) treatments at harvest in 2016 (**A**,**B**) and 2017 (**C**,**D**) experiments. Data are the mean ± SE of quantifications made in duplicate in three replicates. Different letters show significant differences (*p* < 0.05) among treatments for each individual anthocyanin.

**Figure 4 antioxidants-09-00832-f004:**
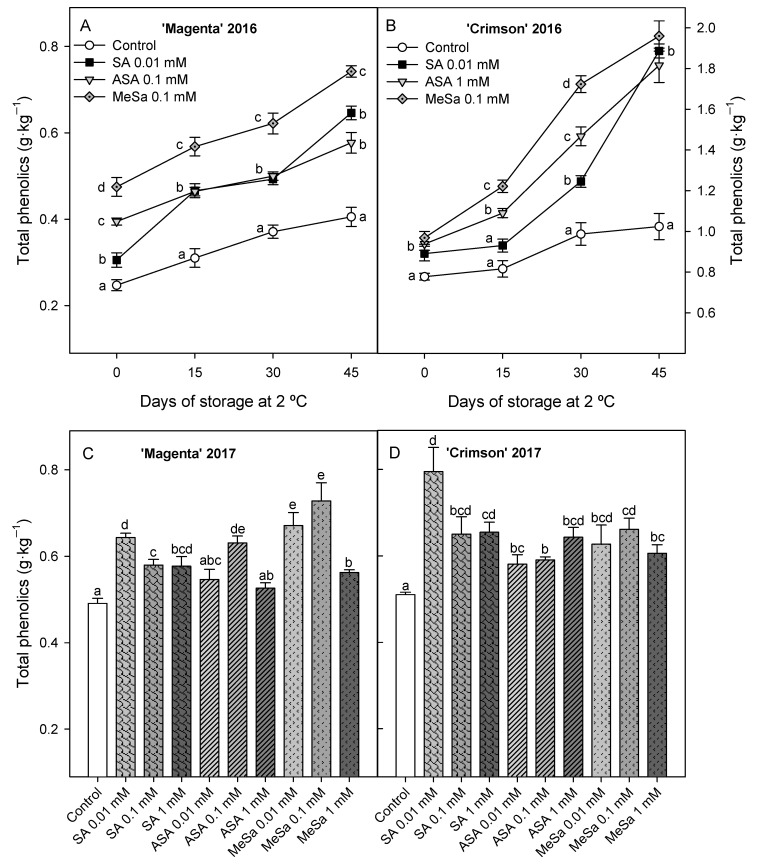
Total phenolic concentration at harvest and during storage at 2 °C in 2016 experiment (**A**,**B**) and at harvest in 2017 experiment (**C**,**D**) in control and salicylic acid (SA), acetyl salicylic acid (ASA) and methyl salicylate (MeSa) treated berries of ‘Magenta’ and ‘Crimson’ cultivars. Data are the mean ± SE of quantifications made in duplicate in three replicates. Different letters show significant differences (*p* < 0.05) among treatments for each sampling date.

**Figure 5 antioxidants-09-00832-f005:**
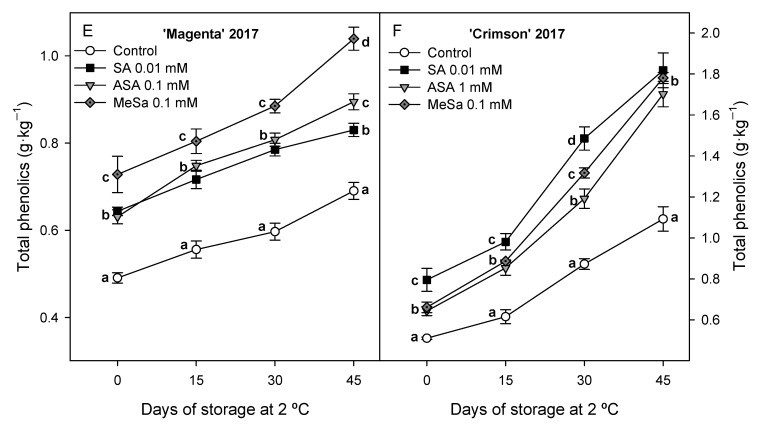
Total phenolic concentration at harvest and during storage at 2 °C in 2017 experiment in control and salicylic acid (SA), acetyl salicylic acid (ASA) and methyl salicylate (MeSa) treated berries of ‘Magenta’ (**A**) and ‘Crimson’ (**B**) cultivars. Data are the mean ± SE of quantifications made in duplicate in three replicates. Different letters show significant differences (*p* < 0.05) among treatments for each sampling date.

**Figure 6 antioxidants-09-00832-f006:**
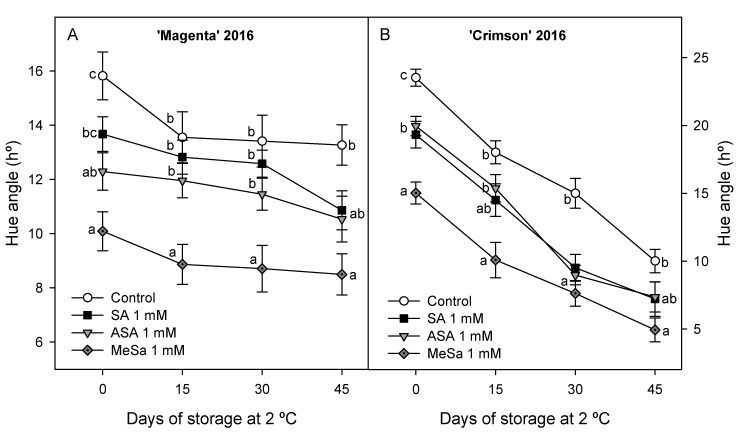
Effects of preharvest 1 mM salicylic acid (SA), acetyl salicylic acid (ASA) and methyl salicylate (MeSa) treatments on Hue angle colour evolution of ‘Magenta’ (**A**) and ‘Crimson’ (**B**) table grapes during storage at 2 °C in 2016 experiment. Data are the mean ± SE of measures made on 30 berries for each of the three replicates. Different letters show significant differences (*p* < 0.05) among treatments for each sampling date.

**Table 1 antioxidants-09-00832-t001:** Dates of salicylic acid (SA), acetyl salicylic acid (ASA) and methyl salicylate (MeSa) treatments (T1, T2 and T3) of ‘Magenta’ and ‘Crimson’ cultivars.

Cultivar	‘Magenta’		‘Crimson’	
Treatment	2016	2017	2016	2017
T1	23rd June	27th June	24th June	28th June
T2	8th July	12th July	9th July	15th July
T3	18th July	21st July	25th July	28th July

**Table 2 antioxidants-09-00832-t002:** Total soluble solids (g·kg^−1^) at harvest (Day 0) and after 45 days of storage at 2 °C (Day 45) in ‘Magenta’ and ‘Crimson’ table grapes as affected by preharvest salicylic acid (SA), acetyl salicylic acid (ASA) and methyl salicylate (MeSa) treatments in 2016 and 2017 experiment.

Cultivar	‘Magenta’ 2016		‘Crimson’ 2016	
	Day 0	Day 45	Day 0	Day 45
Control	195.5 ± 1.0 ^aA^	204.5 ± 3.2 ^aB^	186.3 ± 1.5 ^aA^	204.3 ± 2.1 ^aB^
SA 1 mM	206.1 ± 3.1 ^bA^	214.2 ± 1.5 ^bA^	195.5 ± 2.0 ^bA^	218.2 ± 3.7 ^bB^
ASA 1 mM	207.8 ± 2.9 ^bA^	218.9 ± 3.7 ^bA^	198.2 ± 1.6 ^bA^	223.0 ± 2.3 ^bB^
MeSa 1 mM	207.2 ± 2.7 ^bA^	216.8 ± 2.9 ^bA^	196.4 ± 1.3 ^bA^	217.3 ± 1.7 ^bB^
	‘Magenta’ 2017		‘Crimson’ 2017	
	Day 0	Day 45	Day 0	Day 45
Control	180.2 ± 2.2 ^aA^	191.9 ± 2.5 ^aB^	175.8 ± 1.2 ^aA^	199.0 ± 2.3 ^aB^
SA 0.01 mM	185.5 ± 6.4 ^abA^	198.0 ± 0.5 ^aA^	178.7 ± 0.5 ^abA^	200.3 ± 1.4 ^aB^
SA 0.1 mM	186.5 ± 5.2 ^abA^	201.2 ± 3.2 ^aA^	179.9 ± 2.6 ^abA^	204.0 ± 1.3 ^aB^
SA 1 mM	199.0 ± 2.6 ^bA^	206.8 ± 2.1 ^bA^	181.3 ± 0.9 ^bA^	210.3 ± 1.2 ^bB^
ASA 0.01 mM	186.5 ± 5.6 ^abA^	198.8 ± 1.6 ^aA^	179.5 ± 1.0 ^abA^	193.7 ± 2.4 ^aB^
ASA 0.1	189.3 ± 3.2 ^abA^	191.8 ± 0.9 ^aA^	178.7 ± 3.8 ^abA^	199.7 ± 1.8 ^aB^
ASA 1 mM	205.7 ± 6.9 ^bA^	214.7 ± 3.1 ^bA^	190.3 ± 2.1 ^cA^	214.0 ± 1.6 ^bB^
MeSa 0.01 mM	190.7 ± 4.8 ^abA^	196.7 ± 1.4 ^aA^	178.7 ± 2.0 ^abA^	192.2 ± 1.4 ^aB^
MeSa 0.1 mM	187.2 ± 3.4 ^abA^	195.2 ± 1.4 ^aA^	174.2 ± 5.5 ^abA^	191.5 ± 1.5 ^aB^
MeSa 1 mM	197.8 ± 5.4 ^bA^	207.7 ± 4.2 ^bA^	195.2 ± 2.6 ^cA^	212.5 ± 2.5 ^bB^

Different capital letters show significant differences for each treatment during storage (according to Student *t*’ test) and different lowercase letters show significant differences among treatments for each sampling date (according to ANOVA analysis) at *p* < 0.05.

**Table 3 antioxidants-09-00832-t003:** Total acidity (g·kg^−1^) at harvest (Day 0) and after 45 days of storage at 2 °C (Day 45) in ‘Magenta’ and ‘Crimson’ table grapes as affected by preharvest salicylic acid (SA), acetyl salicylic acid (ASA) and methyl salicylate (MeSa) treatments in 2016 and 2017 experiment.

	‘Magenta’ 2016		‘Crimson’ 2016	
	Day 0	Day 45	Day 0	Day 45
Control	7.5 ± 0.2 ^aA^	6.1 ± 0.2 ^aB^	9.0 ± 0.1 ^aA^	6.6 ± 0.2 ^aB^
SA 1 mM	8.6 ± 0.3 ^bA^	7.0 ± 0.2 ^bB^	11.8 ± 0.4 ^bA^	7.7 ± 0.2 ^bB^
ASA 1 mM	8.5 ± 0.2 ^bA^	6.9 ± 0.1 ^bB^	10.8 ± 0.4 ^bA^	7.6 ± 0.3 ^bB^
MeSa 1 mM	8.7 ± 0.1 ^bA^	7.2 ± 0.3 ^bB^	12.4 ± 0.1 ^bA^	7.8 ± 0.3 ^bB^
	‘Magenta’ 2017		‘Crimson’ 2017	
	Day 0	Day 45	Day 0	Day 45
Control	8.6 ± 0.2 ^aA^	7.3 ± 0.3 ^aB^	10.2 ± 0.1 ^aA^	8.1 ± 0.2 ^aB^
SA 0.01 mM	11.3 ± 0.1 ^cA^	9.6 ± 0.2 ^cdB^	14.1 ± 0.2 ^cA^	12.3 ± 0.3 ^cB^
SA 0.1 mM	10.2 ± 0.4 ^bA^	8.5 ± 0.2 ^bB^	12.9 ± 0.3 ^bA^	10.8 ± 0.1 ^bB^
SA 1 mM	10.3 ± 0.3 ^bA^	8.6 ± 0.3 ^bB^	13.0 ± 0.2 ^bA^	10.9 ± 0.3 ^bB^
ASA 0.01 mM	10.6 ± 0.3 ^bcA^	9.3 ± 0.4 ^bcB^	14.3 ± 0.2 ^cdA^	11.0 ± 0.4 ^bB^
ASA 0.1	14.5 ± 0.7 ^dA^	11.1 ± 0.5 ^dB^	14.2 ± 0.5 ^bcdA^	11.1 ± 0.2 ^bB^
ASA 1 mM	11.0 ± 0.1 ^bcA^	9.1 ± 0.2 ^bcB^	15.0 ± 0.2 ^dA^	12.7 ± 0.4 ^cB^
MeSa 0.01 mM	11.1 ± 0.3 ^bcA^	9.0 ± 0.3 ^bcB^	14.3 ± 0.3 ^cdA^	10.7 ± 0.3 ^bB^
MeSa 0.1 mM	11.4 ± 0.2 ^cA^	9.9 ± 0.2 ^cdB^	14.9 ± 0.2 ^dA^	12.5 ± 0.2 ^cB^
MeSa 1 mM	11.0 ± 0.2 ^bcA^	8.9 ± 0.2 ^bcB^	13.7 ± 0.4 ^bcA^	10.6 ± 0.1 ^bB^

Different capital letters show significant differences for each treatment during storage (according to Student *t*’ test) and different lowercase letters show significant differences among treatments for each sampling date (according to ANOVA analysis) at *p* < 0.05.

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
