# Peer review of "Preharvest Salicylate Treatments Enhance Antioxidant Compounds, Color and Crop Yield in Low Pigmented-Table Grape Cultivars and Preserve Quality Traits during Storage"

_antioxidants, 2020, doi:10.3390/antiox9090832_

Round 1

Reviewer 1 Report

The manuscript deals with the interesting  topic about the influence of salicylate treatments on preservation of quality traits during storage carried out  on two table grape cultivars.

The setting of the experimental trials  is similar to other manuscripts by the same authors concerning other fruit species.

Several explanations and integrations, indicated point by point in the attached Pdf fileS, are requested. A major review is suggested. 

Author Response

Answers to review 1‘comments:

Dear reviewer, thank you very much for your useful comments which have aid to improve our original manuscript. Below you can find an itemed list of your comments and suggestions (in different lines of the original manuscript) and the answer and modification performed in the revised manuscript according to your suggestions. The new information added to the revised manuscript is highlighted in red ink.

- Line 43: Yes, ‘Magneta’ and ‘Crimson’ are table grape cultivars. This information has been added to the revised manuscript in sentence in lines 43-44:

However, ‘Magenta’ and ‘Crimson’ are red and purple skin table grape cultivars, respectively, having low colour development.

- Lines 53-56: The order of these sentences has been changed according to your suggestion, as follow: 

However, the effects of ethephon on colour development are inconsistent and can cause berry softening and the high cost of ABA reduces its practical application. Thus, new researches are needed to find out other cost-efficient preharvest treatments able to induce anthocyanin biosynthesis in table grape. Recently, it has been reported that Methyl jasmonate (MeJa) treatments at 0.1 mM, applied at key points of berry development, accelerated colour evolution due to increased anthocyanin biosynthesis in ‘Magenta’ and ‘Crimson’ table grape cultivars [9].

- Line 67 of the original manuscript: What about legal restrictions on these treatments? Please, take into account this aspect in this paragraph. Moreover, could these treatments influence the sensory traits of treated fruits? Consider also this aspect.

Answer: SA, ASA and MeSA are natural compounds in plants and recognized as GRAS (generally recognized as safe) for the United States Food and Drug Administration (FDA) and previous reports have shown that fruit treatment with salicylates do not impart taste or off-flavour to fruit. The following sentence has been added to the revised manuscript about this issue in lines 75-79:

In addition, SA, ASA and MeSA are natural compounds in plants which have been recognized as GRAS (generally recognized as safe) for the United States Food and Drug Administration (FDA) and previous reports have shown that fruit treatments with salicylates do not impart taste or off-flavour to fruit, although some sensory attributes, such as sweetness and firmness, were increased [15-19]. 

- Line 77: Number of reference for Champa et al. has been added (line 81 of the revised MS).

- Lines 81-82: These sentences have been rephrased in lines 83-86 as follow:

Similar results have been reported by Lo'ay and EL-Boray in SA-treated table grapes of this cultivar during storage at 28 °C [21]. Accordingly, preharvest SA treatments reduced berry weight loss and softening during storage at ambient temperature in ‘Thompson’ table grape cultivar [22].

- Line 82: total antioxidants? in full....

Answer: Yes. This issue has been clarified in the revised manuscript as follow (line 87):

higher total antioxidants (TA) …

- Line 91: over....

Answer: “in” has been changed to “over” in this sentence according to your suggestion as follow (line 95):

were evaluated in these two table grape cultivars over two growing seasons

- Line 97: what about rootstock, and training system? Add more informations.

Answer: More information has been added regarding agronomic conditions according to your suggestions (lines 100-108 of the revised manuscript):

This study was performed with ‘Magenta’ and ‘Crimson’, two different Vitis vinifera L. seedless table grape cultivars, both of them grafted onto Paulsen 1103 rootstocks, during two growing seasons (2016 and 2017). Vines were planted in a sandy soil, at 2.5x3 m, in a commercial orchard in Calasparra (Murcia, Spain). The training system consisted on an overhead trellis at a height of 1.9 m above ground level which was upper and laterally covered with a thread warp net to protect the vines from hail, birds and insects. Vines were irrigated according to their water requirements along the growth cycle by using a programmer drip irrigation system consisting on a drip irrigation line per row with 3 emitters per plant. Fertilizers were applied in the irrigation system and pruning and thinning were carried out according to local cultural practices for table grape.

- Line 100: why did you use different concentrations between years? Justify this decision.

Answer: SA, ASA and MeSa treatments were applied at 10, 5 and 1 mM in 2016 and results showed that in general, vine yield was reduced and the ripening process was delayed with 5 and 10 mM salicylate treatments. Thus, in 2017 experiment lower doses were used in order to avoid these detrimental effects and find out beneficial ones. This issue has been clarified in the revised manuscript as follow (lines 110-111):

since in general, salicylates at 5 and 10 mM decreased vine yield and delayed berry ripening process, as commented in Result section.

- Line 101: what about these chemicals?

Answer: SA, ASA and MeSa were purchased from Sigma-Aldrich (Germany). This information has been added in the revised manuscript (lines 111-113):

Treatments were performed by spraying 1.5 L per vine of freshly SA, ASA or MeSa (purchased from Sigma-Aldrich, Germany) prepared solutions containing 0.5% Tween 20 as surfactant.

- Line 105: 40% of its final volume. what would it be? Specify

Answer: This information has been added (lines 116-117):

when berries had ca 40% of their final volume (@ 1650 and 2100 mm3 for ‘Crimson’ and ‘Magenta’, respectively),

- Line 110: T1, T2, T3 in correspondence of 3 different phenological stages.......

Answer: Yes, the treatments were performed at three phenological stages of berry development, pea stage (ca. 40% of berry final volume), veraison and mature stage (3 days before harvesting).

- Line 114: Specify the thresholds per cv.

Answer: For these cultivars TSS content at ripen stage is 170-180 g kg-1. This information has been added in lines 124-125:

Clusters were harvested when berries reached the commercial ripening stage according to size, colour and total soluble solid content of these cultivars, 170-180 g 100 g-1.

- Lines 128-132: explain this decision

Answer: The following sentence has been added to the revised manuscript to explain this decision (lines 133-134):

Storage experiments were performed when enough number of clusters were harvested which was at the second harvest date for both years and cultivars.  

- Line 128: data not found

Answer: Data of weight loss have been added to the revised manuscript on a new table (Table 1S) and commented in the text as follow (lines 319-324):

Finally, cluster weight losses increased during storage reaching final values of 7.13 ± 0.24 and 9.09 ± 0.36 % in ‘Crimson’ and ‘Magenta’ control table grapes, respectively, in 2016 experiment and significantly (P<0.05) lower values in grapes from salicylate treated vines (Table S1). Accordingly, weight losses during storage were also reduced in 2017 experiment as a consequence of preharvest salicylate treatments, the highest effects being found for 0.1 mM MeSa treatment in both cultivars (Tale S1).

- Line 162: The environmental conditions occurred during the experimental trials in the 2 considered crop seasons have been omitted. This is not correct taking into account how they can influence the fruit growth as well as quality parameters, such as total phenols and anthocyanins. This lack must  be absolutely filled. Moreover, considering the concentration used in both years (1mM), it could  be interesting to compare data between crop seasons.

Answer: Climatic data in experimental field from June to August in 2016 and 2017 experiments have been added on a new table (Table S2). As you suggested, climatic conditions and especially minimum temperature can influence table grape quality attributes, mainly anthocyanin content. This issue has been addressed in the revised manuscript as follow (lines 421-438):

On the other hand, previous studies have shown that climatic conditions, especially high temperatures, have a detrimental effect on colour and anthocyanin accumulation in grapes from veraison to ripening [34-35]. In our experimental conditions, medium and maximum temperatures from July to September, when veraison and ripening occurred in both table grape cultivars, were similar for 2016 and 2017 (Table S2). However, minimum temperatures of July, August and September were lower in 2017 than in 2016 (Table S2) which was related to a higher content on total anthocyanins in control and 1 mM salicylate treated berries for both cultivars in 2017 experiment (Table S3). These results could be attributed to both, lower expression levels of anthocyanin biosynthetic genes and lower activities of anthocyanin biosynthetic enzymes, particularly UDP-glucose flavonoid 3-O-glucosyltransferase (UFGT, a key enzyme in anthocyanin biosynthesis), as reported by Mori et al. [34]. With respect to total phenolic concentration at harvest, higher values were found in control and 1 mM salicylate treated berries in 2016 than in 2017 for ‘Crimson’ cultivar, while the contrary occurred in ‘Magenta’ (Table S3) and in turn a clear relationship with the temperatures from version to ripening cannot been found. In fact, in a recent study with wine grape cultivars, it has been reported that the impact of climatic variables on phenolic content is very complex, since maximum, minimum and medium temperatures as well as rain and water stress along the growing cycle have a different impact on individual phenolic compounds [36] and then further research is required to better understand these relationships.

Three new references (34-36) have been added to the above paragraph.

In addition, a new figure adding data of phenolic concentration during storage in 2017 experiment has been added to the revised manuscript, according to review 2’ comments, and the following paragraph has been added to results section (lines 274-278):

Total phenolic concentration increased during storage in berries from control and treated vines, as was found in 2016 experiment, although values were significantly (P<0.05) lower in control berries than in treated ones (Figure 5 A and B). The highest effect on increasing total phenolic content during the whole storage period was found for 0.1 mM Mesa and 0.01 mM SA treatments in ‘Magenta’ and ‘Crimson’ cultivars, respectively.

- Line 170: please, give a quantification of these parameters.

Answer: Information about these parameters has been added according to your suggestion in lines 182-184 of the revised manuscript:

Clusters were harvested when berries reached their commercial ripening stage based on skin colour (homogeneous purple colour of the berries in a cluster) and TSS content characteristic of these cultivars (170-180 g 100 g-1).

- Line 263: explain the significance.....low value = more coloured?

Answer: The significance of lower Hue value has been added according to your suggestion in sentence in line 289:

showing that treated berries had a deeper purple colour than control ones,

- Line 296: why? In mat and meth are cited, data can be show if they have been recorded.

Answer: Weight loses of cluster have been added to the revised manuscript as Table S1 an commented in lines 319-324:

Finally, cluster weight losses increased during storage reaching final values of 7.13 ± 0.24 and 9.09 ± 0.36 % in ‘Crimson’ and ‘Magenta’ control table grapes, respectively, in 2016 experiment and significantly lower values in grapes from salicylate treated vines (Table S1). Accordingly, weight losses during storage were also reduced in 2017 experiment as a consequence of preharvest salicylate treatments, the highest effects being found of 0.1 mM MeSa treatment in both cultivars (Tale S1).

- Line 298: check the mat and meth (° Brix) and correct

Answer: Total soluble solids have been expressed as g kg-1 along the revised manuscript.

- Line 301: if differences are between day 0 and 45, for each treatment and cv, the Student’ t test analysis  is appropriate  and not Anova. Check and correct.

Answer: This issue has been corrected in the revised manuscript (lines 329-330):

Different capital letters show significant differences for each treatment during storage (according to Student t’ test) and different lowercase letters show significant differences among treatments for each sampling date (according to ANOVA analysis) at P< 0.05.

- Line 306: As above.

Answer: This issue has been corrected in the revised manuscript (lines 335-336): As above.

- Lines 312-313: how about the berry size and cluster conformation? Did you have observed fruit drop? It is important to take under consideration in table cvs. In these type of cvs, more than a cumulative yield/plant it should be more indicative the morphological traits of cluster.

Answer: No berry drop was observed as a consequence of salicylate treatments and morphological traits of clusters was similar in control and treated vines. This information has been added to the revised manuscript in lines 342-345:

No berry drop was observed as a consequence of salicylate treatments and morphological traits of clusters was similar in control and treated vines. Thus, the effects of salicylate treatments at high doses on reducing total yield were attributed to a delay or inhibition on the ripening process since many berries failed to ripen properly and some clusters did not reach the requested commercial quality and were discarded.

- Lines 326-321: It is not so clear. The discussion about the physiological way of action of these compounds must be improved.

Answer: Discussion about this issue has been improved as follow (lines 351-363):

Thus, to accelerate the ripening process and achieve higher prices at market the most appropriate concentration of each salicylate should be established for each particular table grape cultivar. In fact, different effects of preharvest SA treatments on grape ripening have been reported depending on cultivar and applied concentration. Thus, in ‘Flame Seedless’ preharvest treatments with 1.5 and 2 mM of SA hastened berry ripening but 1 mM has not effect [20], while ripening was delayed by 0.72 mM SA in ‘Thompson’ grapes [22]. Accordingly, ripening also was delayed in the wine grape cultivar ‘Syrah’ by 0.72 and 3.6 mM SA foliar spray treatments at veraison stage [25]. Similar results were obtained by preharvest treatment with 4 mM SA on the white table grape cultivar ‘El-Bayadi’ [23] and by 7.2 mM SA injected into berries before veraison [26], the delay being attributed to the antagonist effects of SA on ABA biosynthesis, which is the main hormone implied in the ripening of this non-climacteric fruit. In general, these results and the present ones show that preharvest salicylate treatments at high concentration led to a delay the ripening process in grapes while it could be hastened by lower doses.

- Lines 342-343: data are required.

Answer: Data of the effect of salicylate treatments on berry size have been added, according to your suggestion (lines 375-377):

The increase in yield in ‘Magenta’ was due to an increase on berry volume of 5, 7.6 and 13% as a consequence of 0.01 mM SA, 0.1 ASA and 0.1 MeSa treatments, respectively.

- Line 351: ...to only Crimson.....

Answer: This sentence has been re-written to clarify in the revised manuscript (lines 385-386):

although it is not applicable to all table grape cultivars because no significant increases on yield were observed in ‘Crimson’ for the wide range of SA doses assayed.

- Line 355: Is there a positive correlation? You might do the analysis and show it.

Answer: The correlation analysis between anthocyanin concentration and colour Hue values was performed for both table grape cultivars and years and a negative correlation was found between them, that is to say lower values of Hue angle mean higher anthocyanin concentration. The following sentence ha been added to the revised manuscript (lines 391-395):

In fact, highly negative correlations were found between anthocyanin concentration and colour Hue values for both table grape cultivars and years taking into account data of control and treated berries for all sampling dates during storage (‘Magenta’ 2016: y = -0.0068x + 0.1515, r2= 0.9720; ‘Crimson’ 2016: y = -0.0033x + 0.1338, r2= 0.8567; ‘Magenta’ 2017: y = -0.0103x + 0.2283, r2= 0.9059; ‘Crimson’ 2017: y = -0.0085x + 0.2137, r2= 0.9257).

- Figure S1: add the corresponding letters for identification of graphs (A, B), etc.......

Answer: Letters A, B, C, D for identification of figures have been added to figure legend according to your suggestion as follow:

Figure S1: Accumulated yield of ‘Magenta’ table grape as affected by salicylic acid (SA), acetyl salicylic acid (SA) and methyl salicylate (MeSa) treatments in 2016 (A, C, E, respectively) and 2017 (B, D, F, respectively) experiments. Data are the mean ± SE of three replicates of three vines. Different letters show significant differences (P<0.05) among treatments for each harvest date.

- Figure S2: As above.

Answer: Letters A, B, C, D for identification of figures have been added to figure legend according to your suggestion as follow:

Figure S2: Accumulated yield of ‘Crimson’ table grape as affected by salicylic acid (SA), acetyl salicylic acid (SA) and methyl salicylate (MeSa) treatments in 2016 (A, C, E, respectively) and 2017 (B, D, F, respectively) experiments. Data are the mean ± SE of three replicates of three vines. Different letters show significant differences (P<0.05) among treatments for each harvest date.

Reviewer 2 Report

Followings are my concerns. 1. Where is the Figure 2F? (line 198) 2. What is the reason for the increase of total anthocyanin content during storage? Did you measure moisture contnet? This values should be expressed on dry weight basis. This is same for other analyses. ] 3. The changes in total phenolic of 2017 during storage should be included. 4. What is the relationship between hue angle and anthocyanin content? Please discuss this. 5. Please correct following sentence."Preharvest salicylate treatments at high concentration, 5 and 10 mM, delayed berry ripening and 19 reduced crop yield, while at lower concentrations ripening was accelerated and yield increased." 6. Following sentence is not necessary. "All these healthy effects seem to be connected to the human 39 gut microbiota which is involved in the process of assimilation and bio-transformation of these 40 substances [5]."

Author Response

Reviewer 2:

Dear reviewer, thank you very much for your useful comments which have aid to improve our original manuscript. Below you can find an itemed list of your comments and suggestions and the answer and modification performed in the revised manuscript according to your suggestions. The new information added to the revised manuscript is highlighted in red ink.

Followings are my concerns.

  1. Where is the Figure 2F? (line 198)

Answer: It was a mistake, it should be Figure S2F instead of Figure 2F. This mistake has been corrected in the revised manuscript (line 211).

  1. What is the reason for the increase of total anthocyanin content during storage? Did you measure moisture content? This values should be expressed on dry weight basis. This is same for other analyses. ]

Answer: Weight losses during storage were measured and results added on a new table (Table S1) and commented in the text in lines 319-342:

Finally, cluster weight losses increased during storage reaching final values of 7.13 ± 0.24 and 9.09 ± 0.36 % in ‘Crimson’ and ‘Magenta’ control table grapes, respectively, in 2016 experiment and significantly (P<0.05) lower values in grapes from salicylate treated vines (Table S1). Accordingly, weight losses during storage were also reduced in 2017 experiment as a consequence of preharvest salicylate treatments, the highest effects being found for 0.1 mM MeSa treatment in both cultivars (Tale S1).

Concentration in bioactive compounds as well as total soluble solids and titratable acidy have been expressed in a fresh weight basis according to previous published papers concerning storage experiments on table grapes. However, to show that the increment found in the concentration of phenolic and anthocyanin during storage are not due to their concentration in grape tissues as a consequence of weight loss, the following sentences have been added to the revised manuscript (lines 412-420):

These effects would be even higher after prolonged cold storage since, in general, the highest differences among control and treated berries in total phenolic and total anthocyanin concentrations were found at the last sampling date. As a general trend, 1.5-2-fold increases were found in total phenolic and anthocyanin concentration from harvest to day 45 of cold storage, which cannot be attribute to concentration of the compounds in berry tissues due to weight losses because were lower than 9% in both cultivars and years. Increases in total phenolic and anthocyanin concentrations during cold storage have been reported in other table grape cultivars, such as ‘Flame Seedless’ and ‘Red Globe’ after 45 and 60 days of storage, respectively, in which weight losses were ca. 1% [32-33]. 

Two new references [32-33] have been added to the revised manuscript to support these comments.

  1. The changes in total phenolic of 2017 during storage should be included.

Answer: Data of total phenolic content during storage in 2007 experiment have been included in the revised manuscript (Figure 5) and then, Figure 5 (colour Hue) has been changed to Figure 6. In addition, the following sentences have been added to the revised manuscript to comment these new phenolic added data (lines 274-278):

Total phenolic concentration increased during storage in berries from control and treated vines, as was found in 2016 experiment, although values were significantly (P<0.05) lower in control berries than in treated ones (Figure 5 A and B). The highest effect on increasing total phenolic content during the whole storage period was found for 0.1 mM Mesa and 0.01 mM SA tretments in ‘Magenta’ and ‘Crimson’ cultivars, respectively.

  1. What is the relationship between hue angle and anthocyanin content? Please discuss this.

Answer: A lower value of Hue angle shows a deeper red colour of berries. A brief comment about this relationship had been addressed in the original manuscript (lines 353-355). However, more discussion has been added about this issue in the revised manuscript, and correlation between anthocyanin concentration and colour Hue have been commented, according to your suggestion and those of the reviewer 1 as follow: (lines 391-395):

In fact, highly negative correlations were found between anthocyanin concentration and colour Hue values for both table grape cultivars and years taking into account data of control and treated berries for all sampling dates during storage (‘Magenta’ 2016: y = -0.0068x + 0.1515, r2= 0.9720; ‘Crimson’ 2016: y = -0.0033x + 0.1338, r2= 0.8567; ‘Magenta’ 2017: y = -0.0103x + 0.2283, r2= 0.9059; ‘Crimson’ 2017: y = -0.0085x + 0.2137, r2= 0.9257).

  1. Please correct following sentence."Preharvest salicylate treatments at high concentration, 5 and 10 mM, delayed berry ripening and 19 reduced crop yield, while at lower concentrations ripening was accelerated and yield increased."

Answer: This sentence has been changed in the revised manuscript as follow (line 20):

… while ripening was accelerated and yield increased at lower concentrations.

  1. Following sentence is not necessary. "All these healthy effects seem to be connected to the human 39 gut microbiota which is involved in the process of assimilation and bio-transformation of these 40 substances [5]."

Answer: This sentence has been erased according to your suggestions.

Reviewer 3 Report

The article is out of scope.

Authors could submit the article to suitable journals.

Author Response

Dear reviewer,

We honestly think that the manuscript is appropriate for publication in this special issue since it si focused on the effect of preharvest salicylate treatments on increasing antioxidant compounds in table grape.

The original manuscript has been revised and modified according to your suggestions and those from the other two erviewers.

Thank you very much for your coooments. 

Round 2

Reviewer 1 Report

The manuscript has been improved taking into acoount suggestions and comment. In my opinion, it may be accept for publication.

Reviewer 2 Report

I would recommend this manuscript accept as is.

Reviewer 3 Report

No extra comment.